# Genetic Diversity of *Aspergillus flavus* on Maize in Guatemala

**DOI:** 10.3390/foods12203864

**Published:** 2023-10-21

**Authors:** Mark A. Weaver, Curt Bowen, Lilly C. Park, Angela Bastidas, Samantha G. Drewry, Jennifer R. Mandel

**Affiliations:** 1USDA ARS National Biological Control Laboratory, 59 Lee Road, Stoneville, MS 38776, USA; lilly.park@usda.gov; 2Semilla Nueva, 7 Avenida 14-44 Zona 9 Edificio La Galería, Oficina 35 Guatemala, Guatemala City 01009, Guatemala; curtbowen@semillanueva.org (C.B.);; 3Department of Biological Sciences, University of Memphis, 3774 Walker Avenue, Memphis, TN 38152, USA; sgdrewry@memphis.edu (S.G.D.); jmandel@memphis.edu (J.R.M.)

**Keywords:** mycotoxin, population genetics, simple sequence repeat markers, short tandem repeat, corn, nutrition

## Abstract

Aflatoxin contamination of maize is a leading threat to health in Guatemala. This contamination is the result of infection from *Aspergillus flavus* and has been effectively reduced in other countries through application of nonaflatoxigenic, indigenous strains of *A. flavus*. We collected 82 maize samples from throughout Guatemala in two years and isolated 272 *A. flavus* from these samples, including 126 unique genotypes. We provide here a phenotypic and simple sequence repeat (SSR)-based genotypic description of these isolates, as well as an analysis of the diversity of this population. High levels of genetic diversity were observed with the nonaflatoxigenic isolates in this study, but this information contributes to the development of indigenous aflatoxin biocontrol products.

## 1. Introduction

Maize is central to the lives of people in Guatemala. Over 40% of the agriculture land in Guatemala is devoted to maize cultivation, and about 60% of this is managed by subsistence farmers [1]. Over 900,000 families produce maize, including about 600,000 families in poverty and over 150,000 families in extreme poverty [2]. Annual per capita maize consumption is 92 kg and maize food products are foundational to the diet in rural households [3]. While the best commercial farms might yield over 8 MT/ha (ca. 130 bu/acre) [4], the national average production is about 2 MT/ha and subsistence farmers are only slightly over 1 MT/ha [3].

*Aspergillus flavus* is a minor, opportunistic pathogen of maize, but is highly important because infected maize can become contaminated with aflatoxin (AFs), a set of related secondary metabolites. Aflatoxin B1, one of these metabolites, is the most potent carcinogen found in nature [5]. In high-income nations of the world, the maize trade is well regulated and robustly monitored, and AF-contaminated maize is removed from the market or diverted to lower-value uses (e.g., industrial applications). AF contamination is undetected and unregulated in maize outside of the commercial markets, as is the case with subsistence farming. Factors that facilitate *A. flavus* infection of maize such as extreme heat, drought stress, insect damage, poor fertility, and poor storage conditions are commonplace in Guatemala. Furthermore, several practices that mitigate *A. flavus* infection, including genetically engineered insect resistance, irrigation, appropriate nitrogen fertility, prompt harvest and rapid drying and storage of the crop, are generally out of reach for Guatemalan farmers. This environment and the associated cultural practices support a high incidence of aflatoxin in maize, and dangerous aflatoxin concentrations are frequently observed. One survey in Guatemala noted that 22% of maize samples contained over 20 ppb, the threshold for general consumption in the U.S, and 14% contained over 10 times the threshold (200 ppb) [6]. The people of Guatemala have unhealthy biomarkers consistent with chronic aflatoxin exposure and associated with cirrhosis, liver cancer and childhood stunting [6,7,8,9].

In some maize-growing nations AF contamination of maize has been prevented using a biological control strategy of applying indigenous isolates of *A. flavus* that are aflatoxin non-producers [10,11]. This biological control strategy has been highly effective, typically resulting in over a 90% reduction in AF, and is relatively affordable [12].

Our objectives in the present research were to document the genetic diversity of *A. flavus* on Guatemalan maize samples. Isolates were collected, examined for aflatoxigenicity and genetically characterized with SSR and mating type markers to contribute towards the identification of biocontrol isolates suitable for AF mitigation in Guatemala.

## 2. Materials and Methods

### 2.1. Isolation and Characterization of Aspergillus favus from Grain Samples

Grain samples were collected directly from maize producers in February 2021 and March 2023 and delivered to the USDA-ARS Biological Control of Pests Research Unit, Stoneville, MS, under APHIS permit. Milled grain samples were diluted with 0.1% Triton before being isolated on mDRB media [13], with 4–5 isolates collected per grain sample and 272 isolates collected in total. Isolates from mDRB were transferred to ß-cyclodextrin-adjusted potato dextrose agar (ß-CD PDA) for morphological characterization. Sporulating agar plugs from ß-CD PDA were preserved in sterile water in screw-top vials at room temperature and archived for future research. After 6–7 days of incubation in the dark at 28 °C the undersides of the cultures were observed and the color, ‘white’ or ‘yellow’, was recorded, as described previously [14]. The media was examined under UV light for the presence of fluorescence, which in some studies, has been linked to aflatoxigenicity [14]. The aflatoxigenicity of each isolate was evaluated by growing it on maize, extracting it with 70% methanol and quantifying it using HPLC-FLD, as described previously [15]. They were then grouped as zero, low (1–20 ng/mL), medium (20–300 ng/mL) or high (>300 ng/mL) aflatoxin producers [16,17].

The DNA was isolated from spores using the method outlined by Callicott et al. [18]. Simple sequence repeat (SSR) markers were amplified with 16 primer pairs [16,18,19,20] and amplicons were separated on an ABI 373 DNA Analyzer with the LIZ 500 size standard to genotype *A. flavus* isolates. Strains 21882 and 18543 were also amplified as a quality control/sizing standards. Ramirez-Prado [21] primers were utilized to amplify the *MAT* mating type locus. Geneious Prime 2002.2.1 (www.geneious.com, accessed on 16 October 2023) was used to score SSR and *MAT* locus alleles.

### 2.2. Population Genetic Analyses

The genotypic diversity of the 272 isolates was assessed; however, some of these samples could represent clones and potentially bias genetic diversity statistics. Therefore, we analyzed our data set in two ways: in the first, we assessed haplotypic diversity on the full data set. i.e., 272 samples. In the second, we analyzed genetic diversity on 126 unique haplotypes.

In order to identify the clone-only data set, we used the software program GENODIVEv. 2.0b27 [22] to assign individuals to clones. Assignment to clones was carried out using the Meirmans and Van Tienderen [22] algorithm, which calculates a genetic distance matrix and uses a clonal threshold. This information is used to delineate samples as unique clones using the stepwise mutation model option and missing data is ignored. In the clone-only data set, toxicity groups were assigned to clones by calculating the median for all samples in that clone.

Measures of haplotypic diversity for both data sets were calculated for each SSR locus using GenAlEx 6.5 [23] in Microsoft Excel; number of alleles (Na), number of effective alleles (Ne = 1/(Sum pi2)), Shannon’s information index (I = −1 × Sum (pi × Ln (pi)), haplotype diversity (*h* = 1 − Sum pi2), and unbiased diversity (*uh* = (N/(N − 1)) × h), where N is the number of individuals, pi is the frequency of the *i*-th allele in the population, and Sum pi2 is the sum of the squared population allele frequencies. Significance for differences among toxicity groupings in genetic diversity was assessed using a two-way analysis of variance (ANOVA) with the main effects (factors) of toxicity group and locus, and with the dependent variables (response variables), number of different alleles (Na), number of effective alleles (Ne), information index (I), haploid genetic diversity (*h*), and unbiased haploid genetic diversity (*uh*). Use of a common set of genetic markers across groupings can result in increased statistical resolution for differences among populations because locus-to-locus variation is explicitly included in the model. Two-factor ANOVAs were followed with post hoc LS mean differences Tukey tests using JMP version 13 (SAS Institute, Cary, NC, USA).

The MSTree V2 algorithm was used on the 272 sample dataset with GrapeTree 1.5.0 software to visualize a minimum spanning tree (MSTree). MSTree V2 implements Edmonds’ algorithm [24] and is a novel minimum spanning tree from Zhou et al. [25] and is more tolerant of missing data than classical MSTrees.

## 3. Results and Discussion

### 3.1. Grain Samples and Phenotypes of Isolates

Isolations of *A. flavus* from 34 maize samples in 2021 and 48 maize samples in 2023 resulted in 272 unique isolates. These maize samples included locally produced cultivars; conventional varieties, commercial maize hybrids, as well as recently developed hybrids with improved nutritional qualities [26]. The collection locations are indicated in Figure 1, with detailed location and cultivars in Table 1. The *A. flavus* isolates were scored for pigmentation of the culture media and for the presence or absence of fluorescence when viewed under UV light. Detailed results are available in Appendix A and summarized in Figure 2. There was a nearly even divide among the four visual phenotypes in 2021, and a strong shift in 2023 with over 80% of the isolates being scored as yellow, fluorescent. This shift was associated with an almost complete disappearance of the two white phenotypes.

The aflatoxigenic potential of each isolate was also measured after growth on autoclaved maize. Over half of the isolates in the present survey produced over 20 ppb aflatoxin when grown on maize in laboratory conditions. For comparison, only about 40% of isolates from maize in the United States that was exceptionally highly contaminated with aflatoxin were similarly aflatoxigenic [16]. Less than one-fourth of the isolates were ‘medium’ or ‘high’ aflatoxin producers in another survey of grain isolates from Louisiana, USA [17]. The fact that the Guatemalan *A. flavus* population is shifted so strongly toward a highly toxigenic state is an indication of the difficulty in producing safe maize in that country. It also may represent a surprising opportunity for biological control. That is, if the aflatoxin levels common in Guatemalan maize are related to the highly aflatoxigenic state of the indigenous *A. flavus* population, then shifting that population through application of biocontrol isolates might be especially effective.

### 3.2. Genetic Analysis

The *A. flavus* mating type distribution was different in each of the two years of the present study. As indicated in Figure 3, almost 90% of the *A. flavus* isolated from maize in 2021 was mating type 1–2, while in 2023 there was an even split between *Mat* 1-1 and 1-2. One possible explanation is that the variance is indicative of the small sample size. Alternatively, there are substantial changes in the population from year to year. Some relevant comparisons of population mating type distribution of field collections of *A. flavus* on maize include our work [16], which found 55% to be *Mat* 1-2, and that of Sweany et al. [17] which found 96% of maize isolates to be *Mat* 1-2.

The population genetic data revealed that accounting for clones in the data set indeed influenced the overall patterns of haplotypic diversity (Table 2). We carried out a two-way analysis of variance (ANOVA), which allowed for the use of both gene/locus and toxicity groupings as independent effects (though we were most concerned with difference in toxicity groupings since individual loci may differ in genetic diversity). For the full set of 272 samples, toxicity groupings (zero, low, medium, high) were only significantly different for Na, but this difference went away when the effective number of alleles was calculated (which accounts for sample size), and none of the other measures differed among aflatoxigenicity groupings (Na, Ne, I, *h*, *uh*) (measures summarized in Table 2). Several authors have advocated for accounting for clonality in the data set so as to not bias the results based on abundance or sampling efforts (e.g., [27]); when we accounted for clones in the data set, the genetic differences among these groupings were apparent (Figure 4). The two-way ANOVA revealed that all measures (Na, Ne, I, *h*, *uh*) were significant with respect to the toxicity groupings (significance shown with asterisks in Table 2). The zero grouping, i.e., the nonalfatoxigenic population, had the highest level of genetic diversity across all measures. This presents a challenge for the selection of biological control isolates. The high genetic diversity among nonalfatoxigenic isolates prevents clear identification of especially common, presumably well-adapted genotypes. Overall, measurements of genetic diversity and effective alleles were lower in the present study than a previous data set [16], analyzed with the same alleles and metrics. Critical differences between the two studies are that the previous work included samples from a larger geographic area and from more than two time points.

The minimum spanning tree can be viewed interactively on the web server https://achtman-lab.github.io/GrapeTree/MSTree_holder.html (accessed on 16 October 2023)with the provided profile and metadata files (See Appendix A). The user can view the MSTree by clone genet, toxicity, region, year, etc. One view of the MSTree is presented in Figure 4. In the view presented here, it is apparent that there is aflatoxigenic variation within the clones, indicating that genotypes alone are insufficient in predicting aflatoxigenicity. The most abundant genotype, near the center of this tree, consists of 17 ‘medium’ or ‘high’ aflatoxin producers, consistent with our observation of the highly toxigenic state of this population. Interacting with this MSTree can aid in finding nonaflatoxigenic genotypes that are present in both years or isolated from multiple geographic regions. Interestingly, the genotype of *A. flavus* strain 21882, which has been commercially utilized as a biocontrol product, is present in isolates from both years. Strain 21882 has been thoroughly evaluated as an effective biocontrol isolate [10,28,29], and more recently has been shown to effectively shift the preexisting *A. flavus* population [30]. 

We observed that many of the genotypes were only found in a single region, so we considered the possibility that there could be geographic separation in the populations, i.e., that instead of a single *A. flavus* population, there could be genetic differences from one region to another. Of the total diversity observed in the present collection, the variance attributed to geographic region was only 6%.

## 4. Conclusions and Next Steps

After characterizing 272 *A. flavus* isolates from 82 maize samples, we identified 126 unique genotypes. The calculated diversity of this population is about three times higher than that measured on isolates from commercial corn from Southern US states [16]. The population we isolated from Guatemalan maize was also highly aflatoxigenic, with only 84 nonaflatoxigenic isolates. Differences were found in the colony phenotypes and mating type distribution between the two sample years, so further sampling is needed to document the diversity of this population.

Beyond describing the *A. flavus* population in Guatemala, there is a need to identify biocontrol candidates. The isolates that share a haplotype with strain 21882 merit further examination, but the high diversity of the nonaflatoxigenic isolates makes identification of additional candidates challenging. Biocontrol candidates can be screened for aflatoxin reduction in co-culture with the common, indigenous, highly aflatoxigenic isolates that have been isolated in the present study. Beyond identification of biocontrol strains, challenges remain for the effective and affordable delivery of these strains for aflatoxin mitigation in Guatemala.

## Figures and Tables

**Figure 1 foods-12-03864-f001:**
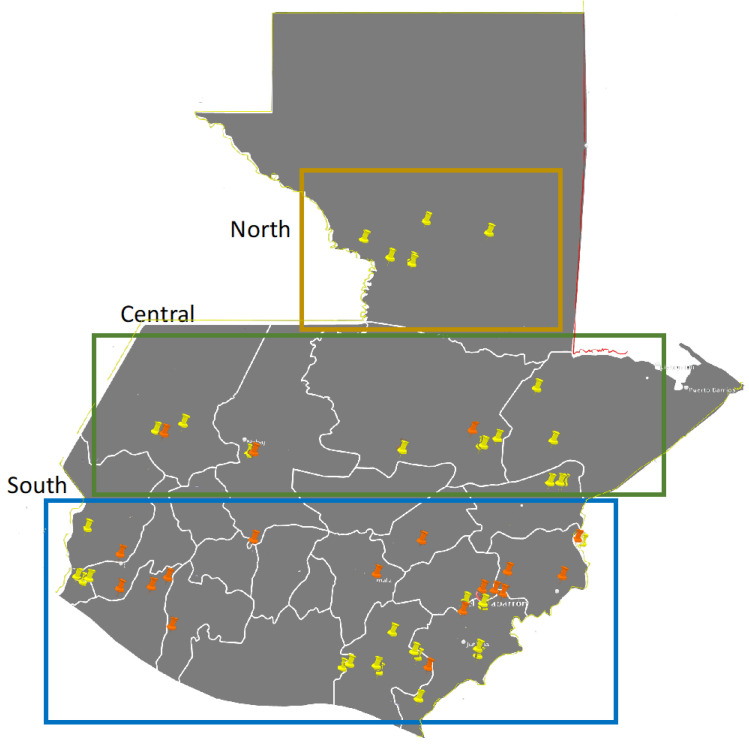
Geographic origin of maize samples. Orange pins indicate 2021 sample points and yellow pins are 2023 sample points.

**Figure 2 foods-12-03864-f002:**
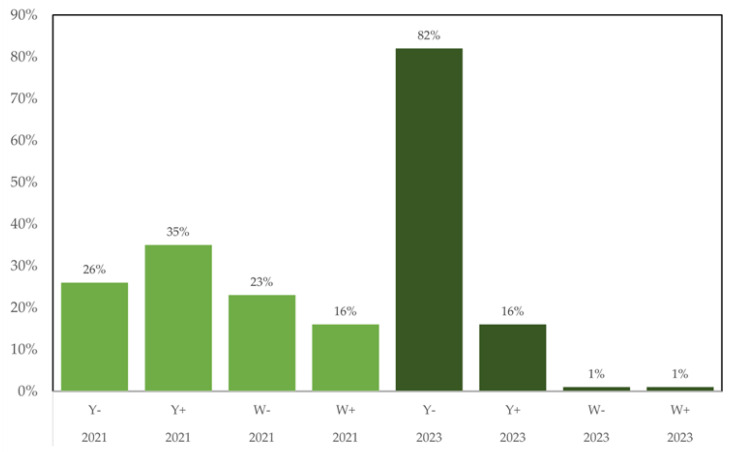
Distribution of phenotypes among A. flavus isolates from maize samples. Y and W refer to yellow and white media pigmentation. The + and − indicate fluorescence under UV light.

**Figure 3 foods-12-03864-f003:**
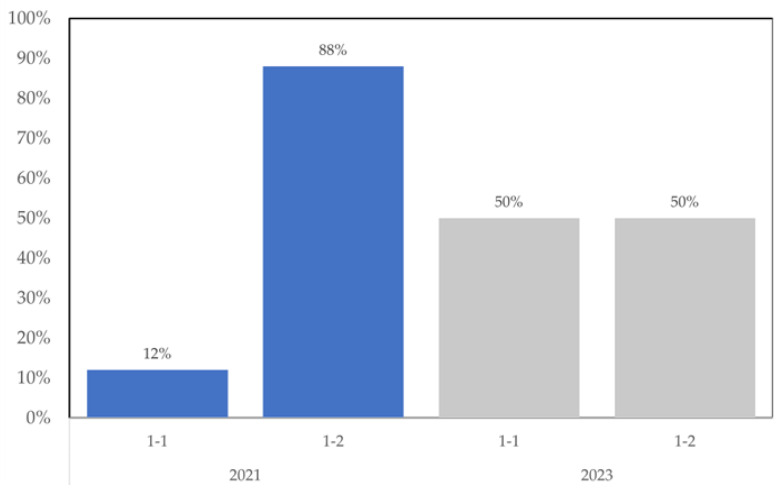
Mating type distribution of *A. flavus* isolates from maize samples.

**Figure 4 foods-12-03864-f004:**
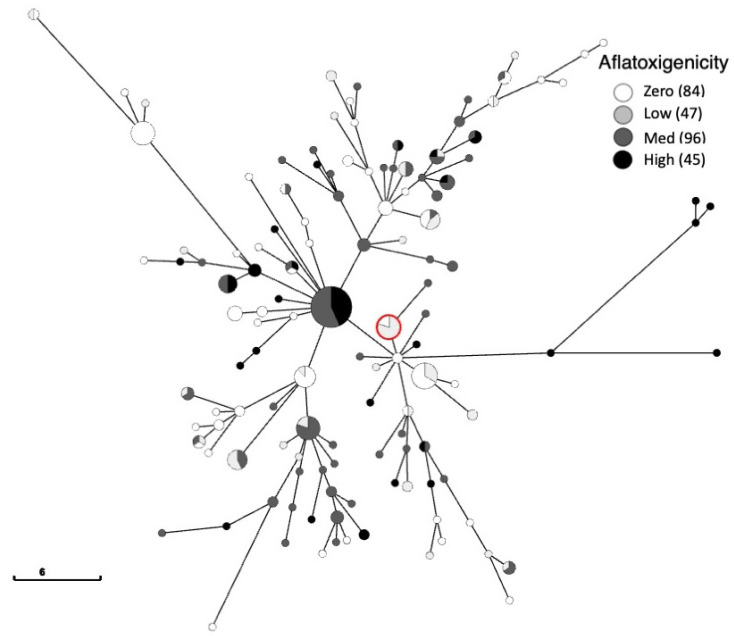
Minimum spanning tree. Bar given for distance scale. Red circle indicates genotypes identical to biocontrol isolate 21882.

**Table 1 foods-12-03864-t001:** Guatemalan maize samples. Cultivar and geographic origin of each sample provided.

**2021 Grain Samples**	**Cultivar**	**Elevation**	**Aldea**	**Municipality**	**Department**
1	HB 83	Midlands	Palos abrazodos	Moyuta	Jutiapa
2	JC-24	Lowlands	Piedra Grande	Chiquimulilla	Santa Rosa
3	JC-24	Lowlands	Piedra Grande	Chiquimulilla	Santa Rosa
4	Fortaleza F3	Lowlands	Placetas	Chiquimulilla	Santa Rosa
5	JC-24	Lowlands	Piedra Grande	Chiquimulilla	Santa Rosa
6	Unknown	Midlands	Obrajuelo	Agua Blanca	Jutiapa
7	Unknown	Midlands		Ipala	Chiquimula
8	Fortaleza F3	Midlands	Poza Verde	S. M. Chaparron	Jalapa
9	H5 SG	Midlands	Las Animas	S. M. Chaparron	Jalapa
10	Fortaleza F3	Midlands		Jalapa	Jalapa
11	Unknown	Midlands	Seamay	Senahu	Alta Verapaz
12	Unknown	Midlands	Paija	S.M. Tucuru	Alta Verapaz
13	Unknown	Midlands	Panhoma	Tamahu	Alta Verapaz
14	Unknown	Midlands	La herradura Chijul	S.M. Tucuru	Alta Verapaz
15	Unknown	Midlands	Peniel	Tacuru	Alta Verapaz
16	Fortaleza F3	Highlands	Payushyec	Sacapulas	Quiche
17	Unknown	Highlands	Pueblo Viejo	San Sebastian	Huehuetenango
18	Unknown	Highlands	Piol	San Sebastian	Huehuetenango
19	Unknown	Highlands	Chumuxuquim	Sacapulas	Quiche
20	Unknown	Highlands	Payushyec	Sacapulas	Quiche
21	DK 410	Lowlands	San Vicente 1	Coatepeque	Quetzaltenango
22	DK 6018	Lowlands	Caballo Blanco	Caballo Blanco	Retalhuleu
23	DK 390	Lowlands	San Rafael	Coatepeque	Quetzaltenango
24	DK 6018	Lowlands	Caballo Blanco	Caballo Blanco	Retalhuleu
25	Unknown	Lowlands	Los Palomos	Los Palomos	Retalhuleu
26	Fortaleza F3	Midlands	Caulotes	Esquipulas	Chiquimula
27	Fortaleza F3	Midlands	Caulotes	Esquipulas	Chiquimula
28	Fortaleza F3	Lowlands	Linea C-8	S. A. Villa Seca	Retalhuleu
29	Fortaleza F3	Lowlands	Linea C-8	S. A. Villa Seca	Retalhuleu
30	Fortaleza F3	Midlands	Olopita	Esquipulas	Chiquimula
31	Fortaleza F3	Midlands	Olopita	Esquipulas	Chiquimula
32	Fortaleza F3	Lowlands	Linea C-18	S. A. Villa Seca	Retalhuleu
33	Unknown	Lowlands	Linea A-3	La maquina	Suchitepequez
34	Unknown	Midlands	El ovejero	Progreso	Jutiapa
**2023 Grain Samples**	**Cultivar**	**Elevation**	**Latitude**	**Longitude**	
1	DK 390	Lowlands	14.05702	−90.36782	Santa Rosa
2	Unknown	Lowlands	14.2781	−90.28683	Santa Rosa
3	HB 83	Lowlands	14.08254	−90.37903	Santa Rosa
4	DK 390	Lowlands	16.58299	−90.09404	Peten
5	HB 83	Lowlands	16.35779	−90.17787	Peten
6	Criollo	Midlands	14.84926	−92.06439	San Marcos
7	Fortaleza F3	Highlands	15.27847	−91.11817	Quiche
8	Criollo	Midlands	14.57462	−92.12152	San Marcos
9	Amarillo	Highlands	15.44222	−91.50228	Huehuetenango
10	DK 390	Midlands	14.56577	−92.05198	San Marcos
11	DK 390	Midlands	14.5472	−92.08604	San Marcos
12	Criollo Blanco	Highlands	15.40105	−91.66328	Huehuetenango
13	NK 102	Lowlands	16.48032	−90.45605	Peten
14	DK 390	Lowlands	16.58241	−90.09541	Peten
15	Criollo	Lowlands	16.37866	−90.30373	Peten
16	Criollo	Lowlands	16.34505	−90.17684	Peten
17	Fortaleza F5	Lowlands	14.07803	−90.57511	Santa Rosa
18	Fortaleza F3	Lowlands	14.09022	−90.37938	Santa Rosa
19	Fortaleza F5	Midlands	13.90407	−90.13655	Jutiapa
20	Fortaleza F5	Lowlands	14.09958	−90.53562	Santa Rosa
21	Unknown	Midlands	15.32417	−89.77944	Alta Verapaz
22	Unknown	Midlands	15.35361	−89.35777	Alta Verapaz
23	Unknown	Midlands	15.32578	−89.75277	Alta Verapaz
24	Unknown	Midlands	16.5175	−89.72888	Alta Verapaz
25	Unknown	Midlands	15.36583	−89.68333	Alta Verapaz
26	Unknown	Midlands	15.30222	−90.23189	Alta Verapaz
27	Unknown	Midlands	15.32861	−89.76222	Alta Verapaz
28	DK 390				
29	DK 390				
30	HS 23				
31	HB 83				
32	Fortaleza F5				
33	Fortaleza F5				
34	HB 83	Lowlands	15.11846	−89.31867	Zacapa
35	HB 83	Lowlands	15.11846	−89.37867	Zacapa
36	Criollo	Midlands	15.64639	−89.45346	Chiquimula
37	Criollo	Midlands	14.44197	−89.76379	Jutiapa
38	Fortaleza F3	Midlands	14.4236	−89.76408	Jutiapa
39	Fortaleza F3	Midlands	14.79819	−89.21239	Chiquimula
40	Fortaleza F3	Midlands	14.78059	−89.19426	Chiquimula
41	DK 390	Midlands	14.16575	−90.15913	Jutiapa
42	DK 390	Lowlands	14.17018	−90.1626	Santa Rosa
43	H-5	Midlands	14.45488	−89.86585	Jalapa
44	H-5	Midlands	14.14853	−89.79325	Jutiapa
45	JC-24	Lowlands	14.16575	−90.15913	Santa Rosa
46	Fortaleza F3	Midlands	14.18684	−89.79283	Jutiapa
47	Fortaleza F3	Lowlands	14.14105	−90.14312	Santa Rosa
48	HB 83	Lowlands	15.119	−89.3	Zacapa

**Table 2 foods-12-03864-t002:** Measures of population genetic diversity and information. Means and standard errors are shown for each population genetic measure: number of individuals sampled (N), number of different alleles (Na), number of effective alleles (Ne), information index (I), haploid genetic diversity (*h*), and unbiased haploid genetic diversity (*uh*). Tukey’s mean separation indicated with letter a–c for each diversity metric.

Aflatoxigenic Status of Isolates	N	Na	Ne	I	*h*	*uh*
Zero	40	5.7 (0.57) a	2.8 (0.46) a	1.1 (0.14) a	0.51 (0.06) a	0.51 (0.06) a
Low	20	2.8 (0.56) b	2.1 (0.32) b	0.61 (0.17) c	0.33 (0.08) c	0.34 (0.09) c
Medium	46	5.1 (0.69) a	2.5 (0.47) a	0.90 (0.17) b	0.42 (0.07) b	0.43 (0.08) b
High	20	2.7 (0.53) b	2.1 (0.35) b	0.61 (0.17) c	0.32 (0.08) c	0.34 (0.09) c
Clone only	126	4.1 (0.33)	2.3 (0.20)	0.80 (0.08)	0.39 (0.04)	0.41 (0.04)
Total	272	7.4 (0.74)	2.3 (0.36)	0.86 (0.15)	0.40 (0.07)	0.40 (0.07)

## Data Availability

The data used to support the findings of this study can be made available by the corresponding author upon request.

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
