# Peer review of "Genetic Diversity of Aspergillus flavus on Maize in Guatemala"

_foods, 2023, doi:10.3390/foods12203864_

Round 1
Reviewer 1 Report
In my opinion the manuscript presents the following issues.
In general, the objectives and rationale for the specific tests carried out do not appear to be presented; In my opinion the term "genetic characterization" is too generic.For example, importance/meaning of the mating type, the analysis of which is described in the materials and methods, is not presented in the introductory part.
It should be made clear from the beginning that/if "biocontrol isolates" means "non-pathogenic A.flavus strains to be disseminated in antagonism to non-pathogenic ones".
(a)The sampling scheme used for the collection of grain specimens is not described.
(b)Number of isolates within the samples is not stated, moreover correlation within the isolates belonging to each sample should be kept into account in the statistical analyses.
(c)ANOVA model should be stated (response, covariates, ...)
(d)There are no tables summarizing the results of analyses. Values for sample number, mean and variability should be given
(a), (b), (c) and (d) could have been shown in supplementary material (that is apparently unavailable to me); in any case, this information, appropriately summarized and presented, should appear in the body of the article.
149) the sentence is not well expressed: different "who from whom"? better: "two (different) mating types distributions were present..."
Figure 4) The metods to generate the curves that appear in the graph are not explained.
Globally, the presentation does not seem well structured; for a tidy and effective presentation of information, I would recommend authors to refer to the guidelines for reporting observational studies, such as, for example, "STROBE" checklist.
In conclusion, I think that the authors should revise the manuscript keeping in mind the above indications.
Author Response
Please see Word file with response to reviewer 1. Also pasted below:
Response to Reviewer 1:
The authors thank Reviewer 1 for the considerable attention to our work. To directly address the questions and concerns raised by Reviewer 1, we have prepared the following point-by-point responses:
In my opinion the manuscript presents the following issues.
In general, the objectives and rationale for the specific tests carried out do not appear to be presented; In my opinion the term "genetic characterization" is too generic.For example, importance/meaning of the mating type, the analysis of which is described in the materials and methods, is not presented in the introductory part.
We have now indicated in the abstract that the “genetic characterization” is SSR-based and noted the dependence on SSRs in the objective statement at the end of the introduction.
It should be made clear from the beginning that/if "biocontrol isolates" means "non-pathogenic A.flavus strains to be disseminated in antagonism to non-pathogenic ones".
It is not accurate to state that any of these isolates are “non-pathogenic”. These isolates differ in aflatoxigenicity, but we have no reason to say they differ in pathogenicity.
(a)The sampling scheme used for the collection of grain specimens is not described.
There is a new Table 1 detailing which samples were collected in which years, with the location and hybrid presented.
(b)Number of isolates within the samples is not stated, moreover correlation within the isolates belonging to each sample should be kept into account in the statistical analyses.
This information is in the Supplemental Table S1. This table contains 272 entries, with 16 SSR entries, mating type phenotypic observations and the source of each isolate. It does not seem possible to include this directly in the manuscript. We have moved a large table into the manuscript to (new Table 1, spanning 2 pages) and there is a new Table 2 with a more coherent summary of the genetic diversity metrics as well as an explicit breakdown of the aflatoxigenicity of the isolates.
(c)ANOVA model should be stated (response, covariates, ...)
The ANOVA model output is in the Supplemental Table 2 (S2)
(d)There are no tables summarizing the results of analyses. Values for sample number, mean and variability should be given
The ANOVA model output is in the Supplemental Table (S2). We have included the summary of this analysis in the new Table 2, replacing the previous Figure 3.
(a), (b), (c) and (d) could have been shown in supplementary material (that is apparently unavailable to me); in any case, this information, appropriately summarized and presented, should appear in the body of the article.
We agree – points (a), (b), (c) and (d) are largely addressed in the previous supplementals, but to make it easier to find, we have moved some of the old ‘supplemental data’ to the main manuscript.
149) the sentence is not well expressed: different "who from whom"? better: "two (different) mating types distributions were present..."
The text indicated in this comment is not in the current draft. I think the new line 148 address this point: “The A. flavus mating type distribution was different in the two years of the present study.” That is to say that the distribution in one year was different thatn the distribution in the other year.
Figure 4) The metods to generate the curves that appear in the graph are not explained.
The ‘curves’ were simply connecting points to make them easier to see. The figure has been replaced by a table.
Globally, the presentation does not seem well structured; for a tidy and effective presentation of information, I would recommend authors to refer to the guidelines for reporting observational studies, such as, for example, "STROBE" checklist.
We were not aware of the STROBE checklist until this reviewer pointed it out. It does raise several interesting questions, but several of them are not directly applicable (we do not have ‘participants’, I do not know what ‘demographic’ characters we can report). Others are addressed. I think the inclusion of the new Table 1, in particular, will answer several of these concerns.
In conclusion, I think that the authors should revise the manuscript keeping in mind the above indications.
Again, our thanks for your substantial input towards the improved presentation of our work.

Reviewer 2 Report
In this study, samples of maize from Guatemala were collected over a two-year period to isolate and described the phenotype and genetic diversity of Aspergillus flavus, and identified candidate varieties of A. flavus for biological control, which has important significance for reducing aflatoxin pollution.
1. As mentioned in the introduction, Aspergillus flavus is a minor, opportunistic pathogen of maize, so what is the main pathogen of corn? Why not choose the main one for follow-up study?
2. The author mentioned in the introduction that “Aflatoxin B1 is the most potent carcinogen found in nature”. Is there no carcinogen as strong or stronger as AFB1? Additionally, the environment and the cultural practices support a high incidence of aflatoxin in maize and dangerous aflatoxin concentrations are frequently observed. Is there a specific number to describe?
3. Lines 153-155 of the results section, some relevant comparisons of population mating type distribution of field collections of A. flavus on maize include our work that found 55% Mat 1-2, it is recommended to cite several references that are consistent with the results.
4. It is mentioned in the materials and methods that aflatoxins were evaluated by HPLC-FLD quantification, and no quantitative results appear to have been seen.
5. The font color and size in Figure 2-5 seem to be inconsistent. Please check and modify it.
6. Line wrap between “2021” in line 150 and “was” in line 152 does not appear to be correct.
7. The position in Figure 1 is inconsistent with others, please adjust it.
8. In lines 10 and 12 of the abstract, the number of spaces after “.” seems too large, and a similar problem exists in the Introduction, Materials and Methods.
9. The doi number format in the references is different, the year positions are inconsistent, and there are many serious formatting errors. Please check and correct them carefully.
Author Response
Please see attached Word file with response to reviewer 2. Also pasted below:
Response to Reviewer 2:
The authors thank Reviewer 2 for the considerable attention to our work. To directly address the questions and concerns raised by Reviewer 2, we have prepared the following point-by-point responses:
- As mentioned in the introduction,Aspergillus flavus is a minor, opportunistic pathogen of maize, so what is the main pathogen of corn? Why not choose the main one for follow-up study?
There are many pathogens that cause greater yield loss in corn, but the economic effect of aflatoxin contamination makes it a major concern. The direct impact of aflatoxin on human health makes it a valid research focus. A Scopus search for “aflatoxin” and “biological control” returned 439 entries within the last 5 years.
- The author mentioned in the introduction that “Aflatoxin B1is the most potent carcinogen found in nature”. Is there no carcinogen as strong or stronger as AFB1? Additionally, the environment and the cultural practices support a high incidence of aflatoxin in maize and dangerous aflatoxin concentrations are frequently observed. Is there a specific number to describe?
AF B1 is the most potent carcinogen found in nature. There are many more potent carcinogens, but not naturally occurring. There are also many natural compounds that are more toxic, but not that are more carcinogenic. We cite the publication of Torres et al (2015) who found 22% of maize samples in Guatemala to be over 20ppb, the U.S. standard for general consumption and about 14% having over 10 times the limit (200ppb). This has been added to the introduction.
- Lines 153-155 of the results section, some relevant comparisons of population mating type distribution of field collections of A. flavuson maize include our work that found 55% Mat 1-2, it is recommended to cite several references that are consistent with the results.
We cited two comparable surveys (ours from 2022 and Sweany, 2017). Other published work on A. flavus mating type is in the context of a population shift in response to applied biocontrol agents (e.g., Molo 2022). We are not aware of others who have published mating types of isolates from a large number of grain samples, especially multiple independent grain samples over a larger geographic area.
- It is mentioned in the materials and methods that aflatoxins were evaluated by HPLC-FLD quantification, and no quantitative results appear to have been seen.
Each isolate was scored for aflatoxigenicity by growing on corn and analysis by HPLC-FLD. The detailed results are in the supplemental table (S1). It is difficult to present these individually in the manuscript (272 isolates, in total), but a summary is given in the new Table 2 and in Figure 4.
- The font color and size in Figure 2-5 seem to be inconsistent. Please check and modify it.
Corrected.
- Line wrap between “2021” in line 150 and “was” in line 152 does not appear to be correct.
Corrected.
- The position in Figure 1 is inconsistent with others, please adjust it.
We have resized and re-positioned the figure.
- In lines 10 and 12 of the abstract, the number of spaces after “.” seems too large, and a similar problem exists in the Introduction, Materials and Methods.
There is some awkward spacing, but this seems to be a function of the justification and the journal template.
- The doi number format in the references is different, the year positions are inconsistent, and there are many serious formatting errors. Please check and correct them carefully.
The reviewer is correct. We have made this correction.
Again, our thanks for your substantial input towards the improved presentation of our work.

Reviewer 3 Report
Manuscript titled "Genetic diversity of Aspergillus flavus on maize in Guatemala and identification of potential biocontrol isolates" presents very interesting topic and warns of a significant threat that Aspergillus flavus and its secondary metabolites present. Searching the literature available online, there are similar papers published, however, those papers have in focus different Aspergillus species and/or different foods and feeds. Therefore, this paper points out new challenges we should tackle.
That being said, there are some corrections needed (listed below):
Abstract section: in my opinion, you should add a few numbers in abstract, meaning, you should mention how many samples were collected, how many were aflatoxigenic, and how many nonaflatoxigenic etc.
Line 31: you wrote "infected maize can become contaminated with aflatoxin (AF), a set of related secondary metabolites.". When written like that, it seems that there is only one aflatoxin, however, A. flavus is able to produce aflatoxins from groups B and G. Therefore, you should write "contaminated with aflatoxins (AFs)".
Line 32: after "Aflatoxin B1", since it is first time mentioned, it should also be written with abbreviation, i.e. "Aflatoxin B1 (AFB1)".
Line 59: You are mentioning MDRB media, but it should be clarified what it is, by mentioning at least full name of medium, that it is modified Dichloran-Rose Bengal medium (mDRB). Also, why use mDRB? Isn't regular DRB or even better, DRBC (Dichloran Rose Bengal Chloramphenicol) agar good enough?
Why are sporulating agar plug stored at room temperature? Unless I'm reading it wrong.
In lines 63-64 you are referring to previously published paper, in which, authors also refer to previously published paper. Therefore, you should ad 3-4 sentences where you briefly describe methods for determination of colony morphology and visual phenotype.
And the same goes for DNA isolation mentioned in lines 68. Briefly describe methods.
Line 73: there is closed parenthesis after web site, however there is no open parenthesis. Also, in this line you mention MAT locus alleles. In my opinion, you should write full name here.
Line 111: You wrote "...and well as recently developed..". Should it be "..as well as recently developed.."?
What kind of criteria is media pigmentation? Why is it important? What does pigmentation of media tell us and what can be concluded by that? Is media pigmentation even a viable phenotypical trait of fungi?
Same goes for fluorescence? What do you detect by it?
Throughout the manuscript, you mention a lot of supplement data. On one occasion, you wrote: "The Minimum Spanning Tree can be viewed interactively on the web server https://achtman-lab.github.io/GrapeTree/MSTree_holder.html with the provided profile and metadata files (See Supplemental files)". In my opinion, there should not be so much data in supplements, instead incorporate it in manuscript.
Lastly, title of your manuscript is "Genetic diversity of Aspergills flavus on maize in Guatemala and identification of potential biocontrol isolates", however, I don't see any results pointing to identification of biocontrol isolates, except mentioning that some nonaflatoxigenic A. flavus are isolated.
As you said it yourself "Biocontrol candidates can be screened for aflatoxin reduction in co-culture with the common, indigenous, highly aflatoxigenic isolates that have been identified in the present study". Maybe rewrite it to "... that have been isolated in the present study", since the screening for biocontroling properties should be done. Also, maybe change the title to "Genetic diversity of Aspergills flavus on maize in Guatemala" since you didn't do any experiments for reduction of aflatoxins in this work.
Some references have DOI, and some don't. All should have DOI at the end.
Considering all of the above, I would recommend that manuscript titled "Genetic diversity of Aspergillus flavus on maize in Guatemala and identification of potential biocontrol isolates" should be reconsidered for publication after major revisions and changes in whole structure. Some areas (as mentioned above) need to be addressed with more details and and information.
English Language is fine and understandable, however, some minor editing should be done. Whole manuscript should be checked for spelling mistakes that can happen.
Author Response
Please see attached Word file with response to reviewer 3. Also pasted here:
Response to Reviewer 3
The authors thank Reviewer 3 for the considerable attention to our work. To directly address the questions and concerns raised by Reviewer 3, we have prepared the following point-by-point responses:
Manuscript titled "Genetic diversity of Aspergillus flavus on maize in Guatemala and identification of potential biocontrol isolates" presents very interesting topic and warns of a significant threat that Aspergillus flavus and its secondary metabolites present. Searching the literature available online, there are similar papers published, however, those papers have in focus different Aspergillus species and/or different foods and feeds. Therefore, this paper points out new challenges we should tackle.
That being said, there are some corrections needed (listed below):
Abstract section: in my opinion, you should add a few numbers in abstract, meaning, you should mention how many samples were collected, how many were aflatoxigenic, and how many nonaflatoxigenic etc.
Thanks. We agree and have added some quantifiable specifics to the abstract.
Line 31: you wrote "infected maize can become contaminated with aflatoxin (AF), a set of related secondary metabolites.". When written like that, it seems that there is only one aflatoxin, however, A. flavus is able to produce aflatoxins from groups B and G. Therefore, you should write "contaminated with aflatoxins (AFs)".
Line 32: after "Aflatoxin B1", since it is first time mentioned, it should also be written with abbreviation, i.e. "Aflatoxin B1 (AFB1)".
It can become awkward to write about aflatoxin / aflatoxins and keeping singular / plural usage consistent and accurate. We agree with this comment and have made this correction.
Line 59: You are mentioning MDRB media, but it should be clarified what it is, by mentioning at least full name of medium, that it is modified Dichloran-Rose Bengal medium (mDRB). Also, why use mDRB? Isn't regular DRB or even better, DRBC (Dichloran Rose Bengal Chloramphenicol) agar good enough?
The capitalization is a bit arbitrary, but the reviewer is correct – The medium was published by Horn and Dorner (1998) who called it mDRB, so we are following their example and have corrected our manuscript. We cannot directly compare mDRB to DRBC because we do not have experience with DRBC.
Why are sporulating agar plug stored at room temperature? Unless I'm reading it wrong.
We are storing them to have a viable culture collection for subsequent research. We store them at room temp in sealed vials at room temperature because this has proven very affordable and effective in our lab for > 4 years.
In lines 63-64 you are referring to previously published paper, in which, authors also refer to previously published paper. Therefore, you should ad 3-4 sentences where you briefly describe methods for determination of colony morphology and visual phenotype.
And the same goes for DNA isolation mentioned in lines 68. Briefly describe methods.
Line 73: there is closed parenthesis after web site, however there is no open parenthesis. Also, in this line you mention MAT locus alleles. In my opinion, you should write full name here.
The open parenthesis is added. We added the name of the locus. Ramerez-Prado refer to it simply as the MAT locus.
Line 111: You wrote "...and well as recently developed..". Should it be "..as well as recently developed.."?
Corrected.
What kind of criteria is media pigmentation? Why is it important? What does pigmentation of media tell us and what can be concluded by that? Is media pigmentation even a viable phenotypical trait of fungi?
Same goes for fluorescence? What do you detect by it?
Abbas et al (2004 and others) indicated that ‘yellow’ pigmentation and ‘fluorescence’ were correlation with aflatoxigenicity. This result has been challenged by others (myself included) who have not seen a similar correlation. The original reports by Abbas et al were based on observations in a very limited geographic area that was dominated by a single genotype that was ‘white, non-fluorescent, nonaflatoxigenic’. Subsequent work with a more diverse population has not supported the conclusions of Abbas et al., 2004. Even if these phenotypes lack the significance originally ascribed to them, they are, at the least, another marker that is easily scored. Including them allows us to maintain some continuity with a large, pre-existing A. flavus collection .
Throughout the manuscript, you mention a lot of supplement data. On one occasion, you wrote: "The Minimum Spanning Tree can be viewed interactively on the web server https://achtman-lab.github.io/GrapeTree/MSTree_holder.html with the provided profile and metadata files (See Supplemental files)". In my opinion, there should not be so much data in supplements, instead incorporate it in manuscript.
We agree. In accordance with your suggestion, we have moved the supplemental table with the source of the maize samples into the manuscript. We are presently include 1 supplemental table as it is likely impossible to include a table with 272 rows and 27 columns in the main text.
In the case of the Minimum Spanning Tree, the ‘result’ is presented in the manuscript as Figure 5. The Grape Tree that is available online allows the user to manipulate the view to better see some other relationships, but the ‘result’ is in the manuscript.
Lastly, title of your manuscript is "Genetic diversity of Aspergills flavus on maize in Guatemala and identification of potential biocontrol isolates", however, I don't see any results pointing to identification of biocontrol isolates, except mentioning that some nonaflatoxigenic A. flavus are isolated.
As you said it yourself "Biocontrol candidates can be screened for aflatoxin reduction in co-culture with the common, indigenous, highly aflatoxigenic isolates that have been identified in the present study". Maybe rewrite it to "... that have been isolated in the present study", since the screening for biocontroling properties should be done. Also, maybe change the title to "Genetic diversity of Aspergills flavus on maize in Guatemala" since you didn't do any experiments for reduction of aflatoxins in this work.
Changed as requested.
Some references have DOI, and some don't. All should have DOI at the end.
Not all references have DOIs, but we have added additional DOIs where possible and corrected the formatting of these entries.
Considering all of the above, I would recommend that manuscript titled "Genetic diversity of Aspergillus flavus on maize in Guatemala and identification of potential biocontrol isolates" should be reconsidered for publication after major revisions and changes in whole structure. Some areas (as mentioned above) need to be addressed with more details and and information.
Again, our thanks for your substantial input towards the improved presentation of our work.

Round 2
Reviewer 2 Report
The revised manuscript has been substantially improved and could be accepted in current form.
Reviewer 3 Report
After corrections in the manuscript, I consider that the paper is ready for publication.